# Preparation, Characterization and Application of Epitaxial Grown BiOBr (110) Film on ZnFe_2_O_4_ Surface with Enhanced Photocatalytic Fenton Oxidation Properties

**DOI:** 10.3390/nano12091508

**Published:** 2022-04-28

**Authors:** Zheng Zhang, Yan Zhang, Zhuo Li, Xueyuan Yang, Xiaolong Yang, Yanhua Peng, Jianqiang Yu

**Affiliations:** School of Chemistry and Chemical Engineering, Qingdao University, Qingdao 266071, China; zhangzheng@163.com (Z.Z.); lizhuo@qdu.edu.cn (Z.L.); yangxy@163.com (X.Y.); yangxl@qdu.edu.cn (X.Y.); yhpeng@qdu.edu.cn (Y.P.)

**Keywords:** bismuth oxybromide, preferential grown, polarization effect, photocatalytic Fenton oxidation, decomposition of tetracycline antibiotics

## Abstract

A novel BiOBr photocatalyst was epitaxially grown in situ onto the surface of ZnFe_2_O_4_, a ferroelectric material with a strong polarization effect. The formatted BiOBr/ZnFe_2_O_4_ composite (BOB/ZFO) showed excellent photocatalytic degradation performance of tetracycline antibiotics (TCs). One of the composites with ZnFe_2_O_4_ content of 10% (BOB/ZFO-10) showed the best properties; the degradation efficiency of TCs upon visible light irradiation for 180 min was 99.2%, which was 3.58 times higher than that of pure phase BiOBr. The functions of ZnFe_2_O_4_ are assumed to be such that the addition of this ferroeletric material not only regulated the spontaneous polarization of BiOBr in the process of synthesis, but also resulted in the construction of Z-scheme heterostructures due to the appropriate staggered band structure of BiOBr and ZnFe_2_O_4_. In the presence of ferroelectric material ZnFe_2_O_4_, the local structure of BiOBr may be distorted accordingly, resulting in preferential growth of a (110) crystal facet of BiOBr and enhancement of spontaneous polarization, which promotes the efficient separation of photogenerated electron-hole pairs of ZnFe_2_O_4_ and BiOBr, and therefore enhances the redox capacity of the photocatalytic degradation of organic pollutants.

## 1. Introduction

As an important product of the pharmaceutical industry, antibiotics have been popularly used in medical care and the breeding industry (livestock breeding and aquaculture), which has unfortunately brought a considerable degree of pollution to the environment. The antibiotics and other emerging pollutants have been detected in rivers, soil and urban wastewater. Due to the wastewater treatment plant being unable to completely eliminate the emerging pollutants [1], they are discharged into the environment, resulting in serious environmental problems. Moreover, in the process of the continuous influence of these antibiotics, there is a significant threat to human health. For example, the prevalence of multiple super pathogenic bacteria in the environment in recent years are mainly induced by the extensive use of antibiotics of human beings. Therefore, wastewater pollution, especially pollution caused by antibiotics, has become an urgent problem to be solved.

These days, the main wastewater treatment processes are based on physical, chemical, physicochemical and biological methods [2,3]. However, when the pollutants in wastewater containing antibiotics need to be removed, these treatment techniques often lose their high efficiency. On the one hand, the biochemical beneficial bacteria might be easily poisoned by the high concentrations of antibiotics and lose their activity when the biological method is used. On the other hand, although the antibiotics are inactivated, the organic toxins that make up antibiotics cannot be completely decomposed. These toxins may still lead to the occurrence of many diseases. Advanced oxidation processes (AOPs), defined as those technologies that utilize the hydroxyl radical (˙OH) for component oxidation [4], have received incredible attention in research on the process of pollution, especially on the degradation of organic as well as inorganic pollutants in wastewater treatment in the past decades. The hydroxyl radical plays an important role in these progresses [5]. However, since the hydroxyl radicals have a very short lifetime, they are produced during application, e.g., by combining the irradiation of ultraviolet light or visible light with a photocatalyst or hydrogen peroxide and metal catalysts (such as Fe^2+^, Cu^2+^) [6]. Fenton and Fenton-like (including photo-Fenton) reactions are well known effective techniques for the removal of various organic contaminants, especially antibiotics [7].

The semiconductor photocatalytic process, as an effective and green technology to solve global energy production and pollution removal in the environment, shows broad prospects for future use [8]. In the past few decades, it has been reported that titanium dioxide, which can only respond to ultraviolet light and accounts for only 4% of the solar light spectrum, showed excellent photocatalytic properties. Therefore, great attention was paid to the applications of titanium dioxide. Unfortunately, expanding the absorbance of visible light by titanium dioxide is still a hard challenge to overcome. So far, various photocatalysts with broad visible-light absorption have been developed, such as bismuth-based materials, silver-based materials, and so on. Of the bismuth-based materials, BiOX (X = Cl, Br, I) and Bi-M-O (M = Mo, V, W) are considered promising photocatalysts [9]. Since a report of high photocatalytic activity of BiOCl in 2006, [10,11,12,13] a variety of applications of BiOX (X = Cl, Br, I) has been reported [14,15,16]. This series of compounds have become promising environmental cleaning materials, which may decompose organic pollutants into non-toxic small molecules with high efficiency [9,17,18,19]. BiOX belongs to a PbFCl crystal structure which has P4/nmm, tetragonal, D4h symmetry [20]. The crystal structure of BiOX can also be regarded as a layered structure, formed by the staggered arrangement of a [Bi_2_O_2_] layer and a double halogen atomic layer [21]. This type of layered structure can be feasibly regulated to be fit for property enhancement [22,23]. BiOBr is the typical BiOX visible light-responsive semiconductor, with a 2.75 eV band gap. Moreover, as a layered semiconductor with an indirect band gap, BiOBr is considered to be able to offer an effective separation capacity of photoinduced electron and hole pairs along the (001) direction [24]. However, the photocatalytic activity of BiOBr is still not applicable for industrialization and needs to be further improved [25,26].

In order to achieve this goal, it is a common strategy to design a composite photocatalyst with a structure that is able to enhance the built-in electric field of the semiconductor and to reduce the recombination probability of photogenerated electrons and holes. MFe_2_O_4_, where M represents a metal cation, which is the typical ferrite with a spinel structure, is stable and has been used in many applications [27], e.g., removal of poison pollution, [28] information storage [29] and pigments [30]. This series of spinel ferrite materials shows a moderate band gap and unique ferromagnetism, which is beneficial for broad light absorption and recovery of the catalyst in the catalytic reaction. Therefore, spinel ferrite materials have great development potential in the field of photocatalysis.

Herein, we design the synthesis of a novel photo-assisted Fenton catalyst BiOBr/ZnFe_2_O_4_ (BOB/ZFO). By the inducing effect of ferroelectric polarization of ZnFe_2_O_4_, the BiOBr crystal grows preferentially along the highly active (110) facet and thus enhances the built-in electric field of BiOBr, which can reduce the rate of carrier-hole pairs recombination. The degradation performance of BOB/ZFO-x (x representing the molar ratio of ZFO) components are improved significantly. The relevant photocatalytic properties and the spontaneous polarization of the mineral materials are further investigated.

## 2. Materials and Methods

### 2.1. Synthesis of BiOBr/ZnFe_2_O_4_ Nanocomposites

All the reagents were purchased from Sinopharm Chemical Reagent Co., Ltd., Shanghai, China and used in the experiments without any further purification. Pure phased BiOBr was synthesized as follows: 2.0 mmol potassium bromide (KBr,)) was dissolved into 60 mL ethylene glycol (EG), while stirring the solution for about half an hour. Then, 2.0 mmol bismuth nitrate (Bi(NO_3_)_3_·5H_2_O) was dissolved into the solution, stirring continuing for another half an hour. After the solution was well prepared, it was heated to 140 °C and kept for 12 h in a hydrothermal autoclave. Finally, the resultant mixture in the autoclave was taken out and washed by water and alcohol three times, and then dried at 60 °C under vacuum overnight.

For the preparation of ZnFe_2_O_4_, 2.0 mmol of ferric chloride and 4.0 mmol of zinc chloride were dissolved into 60 mL of deionized water and stirred for 45 min. After that, the sodium hydroxide was added dropwise to regulate the pH of the solution to 10, the solution charged in a 100 mL hydrothermal autoclave and kept at 180 °C for 24 h. After the hydrothermal reaction was completed, the mixture was taken out, centrifuged and washed with deionized water 5 times. The ZnFe_2_O_4_ powder could be obtained by drying the mixture at 80 °C for 12 h.

The synthesis process of BiOBr/ZnFe_2_O_4_ nanocomposites occurred as follows. A given amount of the abovementioned prepared ZnFe_2_O_4_ powder was dispersed in 60 mL ethylene glycol and stirred for 15 min. Then, 2.0 mmol bismuth nitrate pentahydrate and 2.0 mmol of potassium bromide were added and stirring continued for 30 min, a homogeneous dispersing mixture obtained. After that, the mixture was transferred into a 100 mL hydrothermal autoclave and reacted at 140 °C for 12 h. After the reaction, the product was discharged, washed with deionized water and alcohol 3 times, and centrifuged. Finally, the product was put into a vacuum drying oven and dried at 60 °C for 12 h, and the BiOBr/ZnFe_2_O_4_ nanocomposite photocatalysts with various molar ratios of BiOBr to ZnFe_2_O_4_ (denoted as BOB/ZFO-x) were prepared.

### 2.2. Methods of Characterization

The power X-ray diffraction (XRD) pattern of the as-prepared samples was analyzed by taking advantage of a DX-2700 X-ray diffractometer operating with Cu Kα source (λ = 0.154050 nm) and the *2θ* range varied from 10° to 80°. Field-emission scanning electron microscope (FE-SEM) images were obtained from Sigma 500 (Sigma Corporation, Kanagawa, Japan). Transmission electron microscopy (TEM), high-resolution transmission electron microscopy (HRTEM) and elemental mapping images were taken on a JEM 2200FS. X-ray photoelectron spectroscopy (XPS) measurements were conducted with ESCALAB 250Xi spectrometer (ThermoFisher, Waltham, MA, USA) using Al Kα (hν = 1486.6 eV). The BET surface areas and corresponding pore diameter distribution of bare BiOBr and BOB/ZFO-x hybrid nanocomposites were recorded by a TriStar II 3020 instrument (Micromeritics, Norcross, GA, USA). In order to measure the optical characters of the as-prepared samples, ultraviolet-visible diffuse reflectance spectra (UV–Vis DRS) were conducted with a Hitachi, UH4150 spectrophotometer (Hitachi, Tokyo, Japan). Photoluminescence (PL) spectra were obtained by using a Perkin-Elmer LS 55 (Perkin-Elmer, Waltham, MA, USA) at the excitation wavelength of 350 nm.

### 2.3. Electrochemical Measurements

All the electrochemical measurements were taken on a computer-controlled potentiostat (CHI 660E, CH Instrument Co., Shanghai, China) with a homemade three-electrode system, FTO conductive glass (photocurrent test) with thin-film photocatalyst, Pt wire and Ag/AgCl (KCl, 3.0 mol·L^−1^) used as the working electrode, counter electrode and reference electrode, respectively. The electrochemical impedance spectroscopy (EIS) analysis was carried out by applying an AC voltage amplitude of 5 mV within the frequency range from 10^5^ to 10^−2^ Hz in 0.1 mol·L^−1^ Na_2_SO_4_ solution; Mott-Schottky plots were measured in 0.1 mol·L^−1^ Na_2_SO_4_ aqueous solution by impedance measurement at a fixed frequency of 962 Hz at the applied voltage range of −0.8 V~0.8 V. The photocurrent response was determined in a 0.1 mol·L^−1^ Na_2_SO_4_ aqueous solution by amperometric *i–t* curve measurement with a bias voltage of 0.68 V for 550 s. The light source used in the photocurrent measurement was the same as that used in the photocatalytic measurements in the following section.

### 2.4. Photocatalytic Investigation

The measurements of photocatalytic activity were carried out under the irradiation of a 300 W Xenon long-arc lamp (PLS-SXE300, Beijing Perfectlight Tech. Co., Ltd., Beijing, China) a 420 nm cut-off filter was applied as a visible-light source. In detail, 50 mg of the photocatalysts was dispersed in 50 mg·L^−1^ TCH aqueous solution (50 mL). Then the mixtures were transported into the photocatalytic chemical reaction instrument to be stirred unceasingly for 1 h in darkness to guarantee the establishment of an adsorption-desorption equilibrium between photocatalysts’ surface and pollutant molecules. Subsequently, during visible-light irradiation, 5 mL of suspension was acquired from the reactor at 30 min time intervals, and then the sample was yielded using centrifugation (1000 rpm, 10 min), which was further analyzed by a TU-1810 UV-vis spectrophotometer to record the intensity variations at the characteristic wavelength of 356 nm.

## 3. Results and Discussion

### 3.1. The Crystal Structure of BOB/ZFO Nanocomposite

The crystal structures of the as-fabricated pure BiOBr, ZnFe_2_O_4_ and BOB/ZFO-x composites were analyzed by XRD patterns, the results shown in Figure 1. As can be seen from Figure 1a, a series of peaks at 2*θ* = 10.98°, 21.7°, 25.0°, 31.5°, 32.3°, 39.1°, 46.1°, 50.5°, 56.0° and 57.0° are observed. These diffraction peaks were consistent with the tetragonal phase BiOBr (JCPDS 09-0393), [31] and can be indexed to (001), (002), (101), (102), (110), (112), (200), (202), (114) and (212) lattice facets of pure BiOBr. The sharp and strong peaks indicate a good crystallinity of BiOBr. For the diffraction indexes of BOB/ZFO-x composites, it can be observed that most diffraction peaks are widened and the peak intensity is reduced. This observation might be explained by the grain size of BiOBr crystal becoming smaller due to the coupling of ZnFe_2_O_4_. Moreover, for all BOB/ZFO-x hybrid composites, the obvious diffraction peaks’ index to ZnFe_2_O_4_ cannot be observed. This result might be explained by the small amount of ZnFe_2_O_4_ (less than 15%) or the core-shell structure of BOB/ZFO-x shielding the diffraction of the ZnFe_2_O_4_ crystal.

Figure 1b shows the amplified comparison of the diffraction peaks between bare BiOBr and BOB/ZFO-x hybrid composites. The two typical peaks at 10.98° and 31.5° are attributed to the diffraction patterns of the BOB(001) and BOB(102) crystal facets, respectively. After coupling BiOBr with ZnFe_2_O_4_, the diffraction peaks of these two facets were exposed in pure BiOBr, reduced gradually in intensity with the increasing content of ferroelectric ZnFe_2_O_4_, [32] while the diffraction peaks of BOB(212) and BOB(110) crystal facets were kept invariable. This result is probably explained by the introduction of ZnFe_2_O_4_, with ferroelectricity inducing the preferential growth of BiOBr crystals during the in situ hydrothermal process. This hypothesis is further demonstrated by the diffraction peak shift observed below. Compared with pure BiOBr, the BOB(110) crystal facet in the composites appeared slightly shifted to a larger diffraction angle, which is likely due to the strong interactions between BiOBr and ferroelectric ZnFe_2_O_4_, which result in the distortion of the local structure of BiOBr. It indicates that the intensity ratio of the (102)/(001) facet of BiOBr improves with ZnFe_2_O_4_ percentage increasing to 15% added to the synthesis of BOB/ZFO-x composite.

The variation in the XRD patterns demonstrated that the local structure of BiOBr can be varied by the inducing effect of ferroelectric polarization of ZnFe_2_O_4_ and forming a preferable exposure of the BiOBr (110) facet. As for the role of ferroelectric material on the epitaxial growth of BiOBr, similar work has been reported in our previous paper [33].

### 3.2. The Morphology of BOB/ZFO Nanocomposite

Morphologies of the ZnFe_2_O_4_, BiOBr and BOB/ZFO-x nanocomposites are clearly illustrated by SEM measurements and shown in Figure 2. From Figure 2a, it can be clearly observed that the pure ZnFe_2_O_4_ sample is composed of irregular agglomerated nanoparticles. The agglomeration of ZnFe_2_O_4_ nanoparticles may be caused by the formation of small grains or nanoparticles sticking together due to the ferroelectric magnetism attraction. It indicated that pure BiOBr is formed by particles in an irregular nanosheet structure aggregated together with the size of 100–400 nm. This is the typical morphology of the tetragonal phase BiOBr. As for the BiOBr/ZnFe_2_O_4_ composite, it is interesting that in the presence of ZnFe_2_O_4_, all the BiOBr nanosheets are uniformly covered onto the ZnFe_2_O_4_ surface and exhibit a hierarchical cluster structure, which is composed of the aggregation of many flower-like nanosheets, with particle size around 0.7–2 μm [34]. Moreover, as the ratio of ZnFe_2_O_4_ increases, the flower-like sphere with sharp and well-defined edges becomes indistinct, as shown in the Figure 2d. This observation suggests that the crystallinity of the composite materials decreases with increasing ZnFe_2_O_4_ ratio, which is consistent with the XRD pattern.

Figure 3a shows the TEM image of the BOB/ZFO composite and it is found that the morphology of the material shows a spherical structure, which is consistent with the SEM image. It can be evidently observed that the core of ZnFe_2_O_4_ is covered by a thin BiOBr shell with a distinct interface between two components, indicating that a typical core-shell structure has been formed. This result possibly comes from the fact that due to the ferromagnetic property, the introduced ZnFe_2_O_4_ influenced the growth orientation of BiOBr during the in situ preparation process, which is in agreement with the observation in the XRD characterization. Furthermore, in the HRTEM image of BOB/ZFO-10 composites (Figure 3b), it shows that the lattice spacings of 0.282 nm and 0.278 nm correspond to the (102) crystal facet and the (110) crystal facet of BiOBr, respectively. While the lattice spacing of 0.196 nm corresponds to the (200) crystal facet of ZnFe_2_O_4_ crystal. The HRTEM image of the material was partially amplified and then Fourier transformation was conducted, while several diffraction points were selected for inverse Fourier transform for dislocation analysis (Figure 3d). It was found that the diffraction stripes at different angles appeared blurred as well as distorted. This result ambiguously exhibits the local distortion in the microstructure and the presence of defects. The defects in the lattice tend to provide easier binding sites for the reaction to proceed, thus improving the photocatalytic efficiency. Figure 3e–j shows the elemental mapping of the material. The elements Bi, Br, O, Fe, and Zn were evenly distributed on the whole complex in EDS mapping (Appendix A), indicating the successful synthesis of the BOB/ZFO-x hybrid complex, indicating the presence of ZnFe_2_O_4_ and BiOBr in the composites.

Figure 4a shows the N_2_ adsorption desorption curve of pure BiOBr, ZnFe_2_O_4_ and BOB/ZFO-x composites. It can be seen that all samples showed the typical type IV isotherms with H3 hysteresis loops, which is a typical feature of non-rigid aggregates of flake particles, commonly known as layered pores. Figure 4b shows the pore size distribution, calculated from the adsorption branch of the above isotherms. It reveals that most of the pores for pure phase BiOBr and BOB/ZFO-10 are in the diameter range of 3~60 nm. This result further confirms that the pore distributions of BiOBr and BOB/ZFO nanocomposites are layered pores, which are formed by the aggregation of flake particles, in agreement with the observation in the SEM.

Table 1 gives the surface area and pore size of the pure phased BiOBr and the composite materials. It should be noted that the average pore diameters of BiOBr and ZnFe_2_O_4_ is 9.9 and 8.3 nm, respectively, while those of BOB/ZFO-x nanocomposites increased to 11~13 nm. The increases in the pore diameter results in an increase in the specific surface area of BOB/ZFO nanocomposites. These results offer sufficient evidence for the assumption that the inducing effect of ZnFe_2_O_4_ resulted in the morphology variation of BiOBr. Previous work [35] also demonstrated that the reason why the composite sample owns a large specific surface area is that the ferroelectrical inducing effect of ZnFe_2_O_4_ gives rise to the variation in the morphology of BiOBr. Moreover, the specific surface area of the composites increased with the ratio of ZnFe_2_O_4_, suggesting that the vacancy formed on the surface of the composite increased with the ratio of ZnFe_2_O_4_ added in the synthesis precursor.

### 3.3. The Surface Chemical States of BOB/ZFO-x Nanocomposite

X-ray photoelectron spectroscopy (XPS) was used to detect the surface element state and chemical composition of the as-synthesized pure BiOBr and BOB/ZFO-10 nanocomposites (Figure 5). In Figure 5a, it can be clearly observed the elements of Bi, O, Zn, Fe, and Br are exhibited in the survey spectrum of BOB/ZFO, while the element of C 1s detected at 284.6 eV is used for correcting binding energy. It is worthy to note that no impurity elements were found in the full spectrum of the nanocomposite, indicating the successful preparation of the hybrids’ composite.

In Figure 5b, two splitting peaks, located at 529.8 eV and 530.6 eV, are observed for the BOB sample. They are provided by the Bi-O lattice oxygen and the splitting peak provided by O-H [36,37]. While in the spectrum of BOB/ZFO, the splitting peak provided by O-H is significantly enhanced in intensity and the binding energy is shifted 0.3 eV towards the higher binding energy compared with those in the pure BiOBr. This observation implies that the BOB/ZFO sample gives more surface hydroxyl groups than pure BiOBr, and that the electron migration from BiOBr to Zn_2_FeO_4_ occurring in the BOB/ZFO heterojunction altered the electron density around the atoms. Simultaneously, a new peak located at 533.0 eV, brought by oxygen in ZnFe_2_O_4_, is observed.

As for the spectrum of Br 3d orbitals, the split peaks located at 69.2 eV and 68.2 eV in the pure phase BiOBr, which comes from the binding energies of Br 3d_3/2_ and Br 3d_5/2_ orbitals are observed, respectively (Figure 5c). These two peaks show a negative shift of about 0.3 eV in the spectrum of BOB/ZFO composites. Moreover, Figure 5d exhibits two high-resolution peaks at 164.5 eV and 159.2 eV of Bi 4f spectrum in the pure BiOBr that matched well to the binding energies of Bi 4f_5/2_ and Bi 4f_7/2_, showing the chemical state of Bi element is Bi^3+^, similar to the previous literature. The same negative shift of 0.3 eV binding energy is produced in the BOB/ZFO composite. It is interesting that after coupling with ZnFe_2_O_4_, the binding energies of Br 3d orbitals (69.2 eV and 68.2 eV) and Bi 4f orbitals (164.5 eV and 159.2 eV) in the BOB/ZFO nanocomposites slightly shifted towards the lower binding energy compared with those in the pure BiOBr. The lower binding energy of Br and Bi indicates the electrons tend to transfer from O to Bi and Br, with the electron cloud density of O becoming lower, while the electron cloud density of Br and Bi becomes higher. The above XPS analysis results of the as-synthesized samples showed that the composite was successfully fabricated by a convenient hydrothermal method and there was a strong interaction between ZnFe_2_O_4_ and BiOBr, which was in good accordance with the XRD analysis.

### 3.4. The Local Structure of BOB/ZFO Nanocomposite

Raman spectroscopy is an appropriate technique for probing the local structure of materials because the bonding states in the coordination polyhedra of a material can be deduced directly from the Raman vibrational spectrum. Raman spectra of the synthesized pure BiOBr and BOB/ZFO with various ratios of ZFO were recorded to identify the local structure information (Figure 6). The peaks at 56.5 cm^−1^, 112.0 cm^−1^ and 160.8 cm^−1^ in the spectrum of pure BiOBr are attributed to A_1g_ Bi-Br external stretching, A_1g_ Bi-Br internal stretching and E_1g_ Bi–Br internal stretching, respectively [24,38].

It is well known that Raman shift and half peak width are sensitive to the number of layers in layered materials. Compared with pure BiOBr, several peaks in the composite shifted to lower wavenumbers. As for the BOB/ZFO composite, the two Raman shifts assigned to *A_1g_* internal Bi-Br stretching and the *E_1g_* Bi-Br internal stretching shifted to lower wavenumbers, which can be attributed to the fact that under the inducing effect of ferroelectricity of ZnFe_2_O_4_, intrinsic stresses on the crystal structure might be produced, which therefore alter the periodicity of the lattice and result in the distortion of the local structure. The distortion in the local structure leads to overlap between the Bi 6s and the Br 5p orbital. The greater the degree of distortion of the local structure, the more intrinsic stresses on the crystal structure will be produced, which enhances the migration of photogenerated holes. This result can also be deduced from the results obtained by XRD and XPS.

### 3.5. The Photophysical Property of BOB/ZFO Nanocomposite

The diffuse reflectance UV-Vis spectra (DRS) can be used to determine the band structure of semiconductor materials, from which the photophysical properties of the synthesized materials might be measured. The optical absorption properties of BiOBr, ZnFe_2_O_4_ and BOB/ZFO-x composites were measured by UV-Vis diffuse reflection spectra (DRS) in the wavelength range between 200 and 800 nm. As shown in Figure 7a, all the samples of BiOBr, ZnFe_2_O_4_, BOB/ZFO-1, BOB/ZFO-5, BOB/ZFO-10 and BOB/ZFO-15 exhibited favorable absorbance of light from ultraviolet to visible light. The absorption edge of BiOBr is 430 nm, which is in agreement with that previous reported. However, two absorption edges for pure phased ZnFe_2_O_4_ at about 610 nm and 850 nm can be observed. This indicates that ZnFe_2_O_4_ is capable of absorbing a broad spectrum of solar light, particularly visible light. It is worthwhile to note that the absorbance edge of the BOB/ZFO-x hybrid heterojunction showed a remarkable redshift with respect to pure BiOBr and a blueshift against pure ZnFe_2_O_4_.

This result can also be verified from the following band energy (*E_g_*) of as-fabricated BiOBr, ZnFe_2_O_4_ and BOB/ZFO-x determined by the Kubelka-Munk formula:$${\left(\alpha h\upsilon \right)}^{1/2}=A\left(h\upsilon -E_{g}\right)$$
where hυ is the energy of the incident photons (eV), $E_{g}$ is the optical band gap (eV), and α is the measured absorption coefficient.

The Tauc fit plots (Figure 7b) tell us that the band gap of BiOBr and ZnFe_2_O_4_ are 2.79 eV and 2.13 eV, respectively. The BOB/ZFO-x intercepts of the composites in the *x*-axis are all around 1.9 eV, which exhibited large narrowing bandgaps compared to that of BiOBr and close to that of pure ZnFe_2_O_4_. ZnFe_2_O_4_ is a multiband semiconductor, the bottom of the conduction band (*E*_c_) and middle narrow bands (*E*_m_) are mainly constituted by the Zn 3d and Fe 3d orbitals, respectively, while the top of the valence band (*E*_v_) mainly consists of the O 2p orbital.

The above observations indicate that the BiOBr shell is closely adhered to the surface of ZnFe_2_O_4_ nanoparticles and form a *Z*-scheme heterostructure between ZnFe_2_O_4_ and BiOBr. When two semiconductors (herein, BiOBr and ZnFe_2_O_4_) with appropriate staggered band structures construct a *Z*-scheme heterojunction, the Fermi levels are aligned after contact. The semiconductor with a larger work function (more negative Fermi level, BiOBr herein) accepts the electrons from the counter one with a smaller work function (higher Fermi level, ZnFe_2_O_4_ herein) and results in the downward shift of the band bending. The greater degree of the distortion in the local structure, the more intrinsic stresses on the crystal structure will be produced, which leads to the increase in the band bending and, thus, a decrease in the energy band. Therefore, a space charge region will be formed at the interface of the heterostructure. The consequently generated built-in electric field can promote the separation of photogenerated electrons and holes, and greatly affect its photoelectrochemical properties.

### 3.6. The Photoelectrochemical Properties of BOB/ZFO Nanocomposites

Generally, the separation, transmission, and recombination of photoinduced charge carriers were considered the key influence factors in determining photocatalytic performance [39,40]. Therefore, the photoelectrochemical characterizations of BiOBr, ZnFe_2_O_4_ and BOB/ZFO-x nanocomposites were investigated by transient photocurrent−time spectra (*I–t*) and electrochemical impedance spectroscopy (EIS). The results are shown in Figure 8a and Figure 8b, respectively. As described in Figure 8a, pure BiOBr and ZnFe_2_O_4_ displayed inferior intensive photocurrent due to the rapid recombination of their photogenerated carriers. After BiOBr nanosheets were closely adhered to the surface of ZnFe_2_O_4_ nanoparticles, the intensities of the photocurrent responses of BOB/ZFO-x was considerably improved compared to those of pure BiOBr and ZnFe_2_O_4_, indicating that the core-shell heterojunction structure accelerated the separation and transmission of interface photoelectron-hole pairs, with the highest photocurrent intensity of composites being 26.2 times that of BiOBr. This result is not only attributed to the formation of a heterojunction between BiOBr and ZnFe_2_O_4_, but also due to the distortion of BiOBr local structure induced by ZnFe_2_O_4_ during the synthesis process.

Moreover, the electrochemical impedance spectrum (EIS) is expressed as a Nyquist plot, as shown in Figure 8b. The semicircle diameter of the BOB/ZFO photocatalyst gradually changes as the ZnFe_2_O_4_ ratio varies, and the smaller the Nyquist radius in the EIS, the better the charge transfer capability, which inhibits the recombination of photogenerated carriers. The impedance data were fitted using a fully equivalent circuit. BOB/ZFO composite photocatalysts all have significantly lower charge transfer resistance than that of pure phase BiOBr, where BOB/ZFO-5 has the lowest impedance, followed by BOB/ZFO-10, both of which have higher charge transfer efficiency.

### 3.7. The Photo-Fenton Degradation Property of BOB/ZFO Nanocomposite

In order to further demonstrate that BOB/ZFO-x nanocomposites display noticeably enhanced capability compared with pure phased BiOBr and ZnFe_2_O_4_, TCH degradation efficiency in the visible-light condition (λ > 420 nm) was investigated. In Figure 9a, it can be observed that without photocatalyst and without addition of H_2_O_2_, the degradation efficiencies of TCH are very low. When a certain amount of H_2_O_2_ is supplied to the reaction system, TCH can be degraded even though it is operating in the dark state. These observations demonstrate that the degradation of TCH is the typical Fenton oxidation reaction. Therefore, the number of electrons transferred to H_2_O_2_ with generation of hydroxyl radicals with H_2_O_2_ is the dominant factor for the degradation reaction. The photocatalyst with an enhanced converting efficiency will exhibit excellent TCH degradation performance.

This presumption can be verified by the following degradation investigations. It was found that the degradation efficiencies of TCH over most of the BOB/ZFO-x nanocomposites increased remarkably compared with BiOBr and ZnFe_2_O_4_. Especially, the BOB/ZFO-10 sample manifested up to 99.2% of removal efficiency for TCH in 3 h, which was 3.58 and 1.1 times as large as those of pure BiOBr (27.7%) and ZnFe_2_O_4_ (90.3%), respectively. Moreover, the TCH degradation kinetics were further investigated and the corresponding results all fitted well with the pseudo-first-order model. As shown in Figure 9b, the corresponding rate constant (k) values of BiOBr, ZnFe_2_O_4_ and BOB/ZFO-1, BOB/ZFO-5, BOB/ZFO-10 and BOB/ZFO-15 nanocomposites are 0.133, 0.463, 0.41, 0.622 and 0.38 h^−1^, respectively. Notably, the BOB/ZFO-10 sample exhibited the highest k value, which is about 4.68 and 1.34 times higher than that of pure BiOBr and ZnFe_2_O_4_. The above experimental results indicated the successful coupling of the BiOBr nanosheet and ZnFe_2_O_4_. Nanoparticles for the preparation of BOB/ZFO core-shell heterojunction decrease space charge migration distance and accelerate the separation of electron and hole, thus enhancing the photocatalytic efficiency. We have compared the degradation efficiency of this work and other recent work [25,26,41,42,43,44,45,46]. The comparison of all conditions and degradation rates are shown in Appendix A of the Appendix A. It can be found that despite differing amounts of catalyst, reaction conditions and initial concentration of antibiotics being used, the experimental results in our work show a relatively high reaction performance.

Furthermore, it is well known that the determination of the main active species in the process of the photocatalytic reaction is extremely important in exploring the transfer route of photogenerated electrons and holes. Firstly, according to our previous research experience, ˙O_2_^−^ is the main active species generated in the heterojunction. While at higher pH condition, the built-in electric field from ZnFe_2_O_4_ to BiOBr at the interface of BOB/ZFO nanocomposite will be enhanced, resulting in the generation of more ˙O_2_^−^. Meanwhile, for the photocatalytic Fenton oxidation reaction, the hydroxyl radical is the main oxidative species in the decomposition of refractory organic compounds. The hydroxyl radical is generated through the following formula:H_2_O_2_ + ˙O_2_^−^→˙OH + OH^−^ + O_2_

Therefore, it is worthwhile exploring the optimal pH condition used in the reaction system to produce either an enhanced built-in electrical field, or to make the above equilibrium forward to right, so as to promote the generation of more hydroxyl radicals and enhance the photocatalytic efficiency of TCH degradation. As shown in Figure 9c, the degradation efficiency of TCH is much lower when the reaction is carried out at pH = 5, with only 69.5% of TCH decomposed after 3 h irradiation. However, when the reaction is carried out at pH = 7, almost 99% of TCH can be degraded at 3 h, corresponding to the rate constant (k) values of 1.49 h^−1^, which is 4.38 times that at pH = 5. On the contrary, the degradation efficiency of TCH is much higher at low pH condition: 99.2% of TCH is degraded at 3 h, with a rate constant (k) value of 1.72 h^−1^. These observations demonstrate that the pH condition of the reaction system plays a vital effect in the degradation efficiency of TCH. This may be explained by the generation efficiency of the active species for the oxidation of TCH depending to a considerable extent on the pH value. At the neutral or weak alkaline condition (pH = 7 in this study), the structure of the catalyst is at the optimum status, which is beneficial for the generation and transmission of photogenerated electrons and holes, so that more active species are able to reach the surface of the photocatalyst to participate in the oxidation reaction. On the contrary, at low pH condition (pH = 3 in this study), the Fenton effect may become the main influencing factor for the decomposition of TCH. As is known, the traditional Fenton reaction usually occurs under acidic conditions, and higher concentration of H^+^ ions may promote the reaction of H_2_O_2_ with ˙O_2_^−^ to generate ˙OH species, which can be used for the degradation of organic pollutants.

Finally, the reusability and the stability of the BOB/ZFO-10 composite employed in the photocatalytic Fenton oxidation process was discussed though a recycling experiment. It was clearly observed that after the fifth degradation cycle (Figure 9d), the decomposition percentage of TCH over the BOB/ZFO-10 sample did not remarkably reduce, and still accounted for more than 70% of the original removal rate after the last cycle. These results indicate that the stability of the composite photocatalyst is guaranteed.

ESR measurements can offer information to determine the reactive radicals in the photocatalytic system. The results are shown in Figure 10. It is clear that no obvious ESR signals of DMPO − ˙O_2_^−^ and DMPO − ˙OH were observed in the darkness condition. As can be seen from Figure 11, after 10 min irradiation of visible light, both signals of ˙O_2_^−^ and·OH were detected by BOB/ZFO-10. Since the E_CB_ of ZnFe_2_O_4_ is more negative than the redox potential (−0.33 eV) of O_2_/˙O_2_^−^, the photogenerated electrons in the CB of ZnFe_2_O_4_ can reduce O_2_ to produce ˙O_2_^−^. For ˙OH, the valence band of BiOBr is negative to the redox potential of H_2_O/˙OH (2.38 eV), and photogenerated holes in the valence band cannot oxidize H_2_O to ˙OH. In fact, ˙OH was provided by photo-Fenton reaction. The formula is as follows:H_2_O_2_ + Fe^2+^ →˙OH + OH^−^ + Fe^3+^
Fe^3+^ + H_2_O_2_→Fe^2+^ + ˙OOH + H^+^
˙O_2_^−^ + TCH →→→ H_2_O + CO_2_
˙OH + TCH→→→H_2_O+CO_2_

In the self-assembly process of BiOBr, the ferroelectric ZnFe_2_O_4_ plays roles not only as a crystal attachment point, but also in the intrinsic ferroelectromagnetism affecting the surface morphology [47], crystal facet growth and the direction of the internal polarization field of BiOBr during the material assembly, as in Monzon‘s work [48]. In other words, when BiOBr is synthesized by this process, during the self-assembly process, instead of tending to be a nanosheet structure, the crystal formation of BiOBr is induced by a weak magnetic field around ZnFe_2_O_4_ and a spherical structure with less surface energy is formed around the ZnFe_2_O_4_ crystalline nucleus. Simultaneously, the anions and cations are migrated according to the distribution of the magnetic field, forming a uniformly oriented polarization field with orientation (010). As shown in Figure 11a, the uniformly oriented polarization field provides a photogenerated electron highway (PGEH) for the transfer of electrons. The electrons passing through the PGEH exhibit a higher transfer efficiency and a lower recombination probability. This result agrees well with the significant increase in the photocurrent as well as a short Nyquist radius (Figure 8).

Moreover, as shown in Figure 11b, both BiOBr and ZnFe_2_O_4_ are visible light-responsive photocatalysts and can be excited by visible light to generate electron–hole pairs. The electrons can be transferred to the conduction band and the holes are left on the valence band. If the charge transfer path of photogenerated electron–hole pairs is similar to a typical type II heterojunction system, the photogenerated electrons in the CB of BiOBr will produce fewer ˙O_2_^−^ radicals due to their low reducibility. Therefore, photogenerated electrons in the CB of BiOBr tend to transfer and recombine with photogenerated holes in ZnFe_2_O_4_. In this way, the photogenerated electrons accumulated in the CB of ZnFe_2_O_4_ can be used to activate the adsorbed O_2_ to form more ˙O_2_^−^. At the same time, the photogenerated holes left in the VB of BiOBr directly oxidize the organic pollutants. Since the Fermi energy level of ZnFe_2_O_4_ is higher than that of BiOBr, when ZnFe_2_O_4_ and BiOBr are in close contact, the Fermi energy level difference drives the interface electrons to transfer from ZnFe_2_O_4_ to BiOBr until the Fermi energy level equilibrium is reached. At the interface of the BOB/ZFO nanocomposite, the energy band of ZnFe_2_O_4_ tends to upwardly bend, while the band of BiOBr bends reversely, resulting in a positively charged ZnFe_2_O_4_ and a negatively charged BiOBr. A built-in electric field directed from ZnFe_2_O_4_ to BiOBr is formed. The potential barrier caused by energy band bending hinders the further flow of electrons from ZnFe_2_O_4_ to BiOBr, thus avoiding the formation of type II heterojunction. The built-in electric field makes the photogenerated electrons in the CB of BiOBr combine with the holes on the VB of ZnFe_2_O_4_, which accelerates the separation of photogenerated electrons in the CB of ZnFe_2_O_4_ from the holes on the VB of BiOBr that show high redox, which effectively improves the ability of photocatalytic degradation of pollutants.

## 4. Conclusions

In summary, the BiOBr/ZnFe_2_O_4_ nanocomposite photocatalysts have been synthesized by epitaxial growth of BiOBr(110) crystalline facets on the ferroelectric ZnFe_2_O_4_ surface. The BOB/ZFO-10 composites exhibit the most efficient photocatalytic degradation performance for tetracycline hydrochloride under visible-light irradiation. A degradation degree of 99.2% in 3 h is observed, which is 3.58 times higher than that of pure BiOBr. The enhanced photocatalytic performance of the BiOBr/ZnFe_2_O_4_ composite can be attributed to the establishment of a built-in electrical field at the interface between ZnFe_2_O_4_ and BiOBr, due to the regulation of ferroelectricity of ZnFe_2_O_4_. This resulted in the preferential growth of a (110) crystal facet of BiOBr and enhancement of the spontaneous polarization. This effect promotes the efficient separation of photogenerated carrier-hole pairs of BiOBr. Meanwhile, the ESR spectrum indicated that electrons, ˙O_2_^−^ and ˙OH, played an important role in the photo-Fenton degradation of TCH. The photodegradation performance was significantly improved. Future research should be focus on the study of local structure and clarify the catalytic mechanism in more detail.

## Figures and Tables

**Figure 1 nanomaterials-12-01508-f001:**
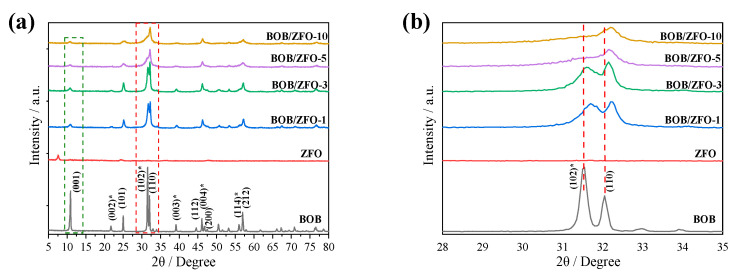
(**a**) XRD patterns of BiOBr, ZnFe_2_O_4_ and BOB/ZFO-x nanocomposites; (**b**) amplified XRD patterns of the samples. (The peaks with a “*” in (**a**) indicated that these peaks in the BOB/ZFO-x are decayed obviously compared with pure BiOBr. This is due to the induced effect of ferroelectric ZnFe_2_O_4_, which makes the (110) facet epitaxially grown, resulting in the decrease in the intensities of these peaks.

**Figure 2 nanomaterials-12-01508-f002:**
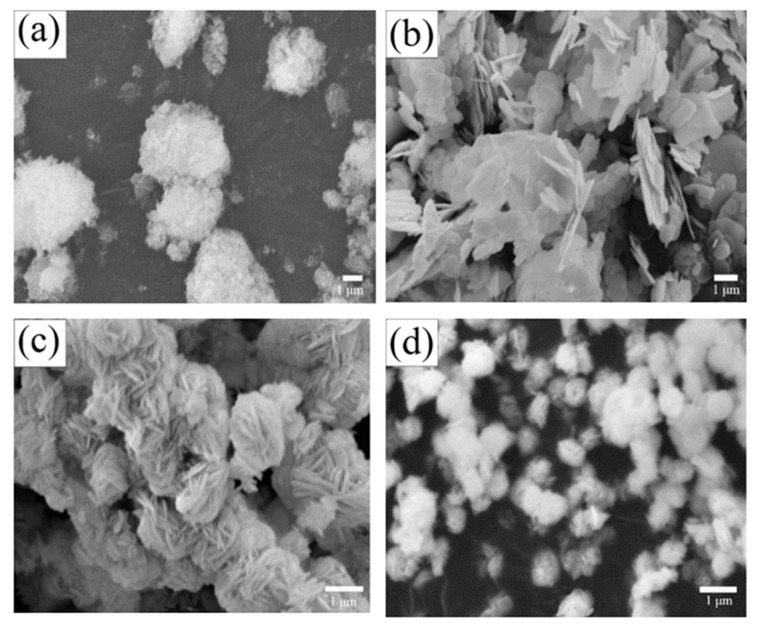
SEM photographs of (**a**) ZnFe_2_O_4_, (**b**) BiOBr, (**c**) BOB/ZFO-5, (**d**) BOB/ZFO-15.

**Figure 3 nanomaterials-12-01508-f003:**
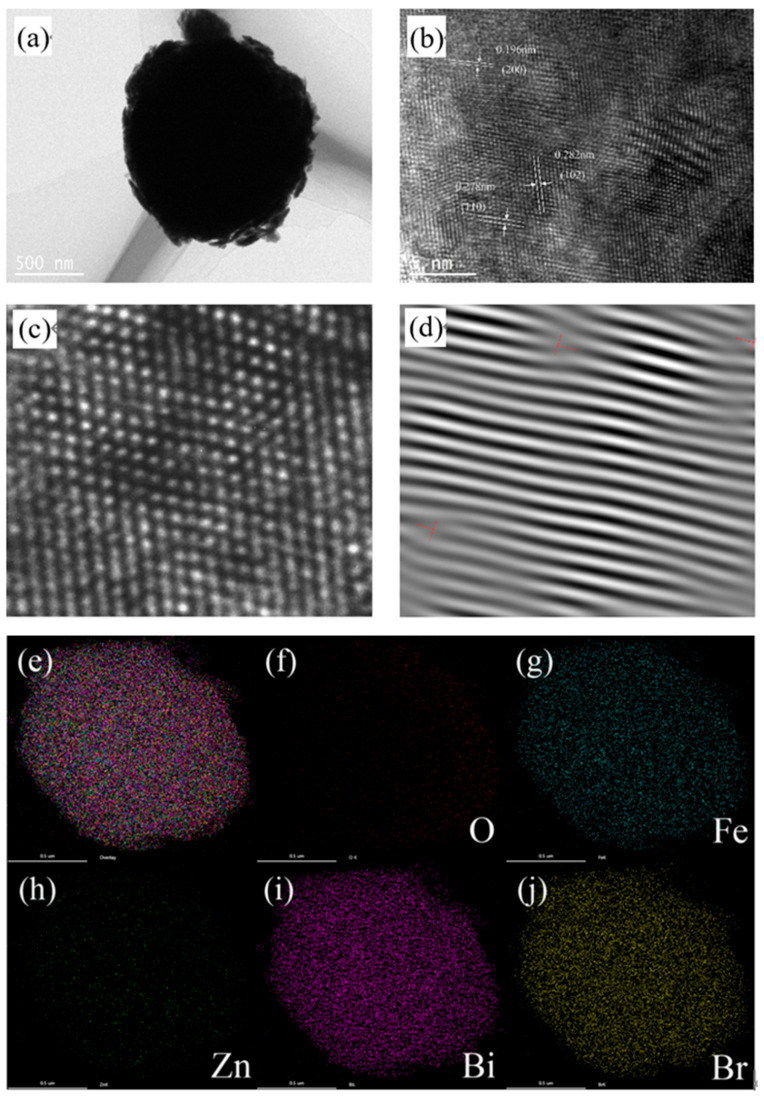
TEM images of (**a**) BOB/ZFO-10 heterojunction; (**b**,**c**) HRTEM images; (**d**) FFT transform image; and (**e**–**j**) element mapping images.

**Figure 4 nanomaterials-12-01508-f004:**
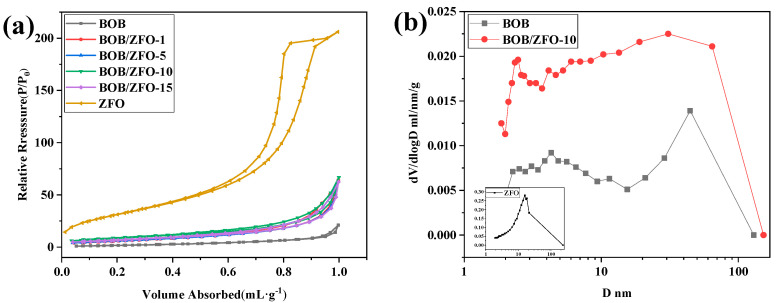
(**a**) Nitrogen adsorption–desorption isotherm and (**b**) the corresponding BJH pore size distribution of the BiOBr, ZnFe_2_O_4_ and BOB/ZFO-x composites.

**Figure 5 nanomaterials-12-01508-f005:**
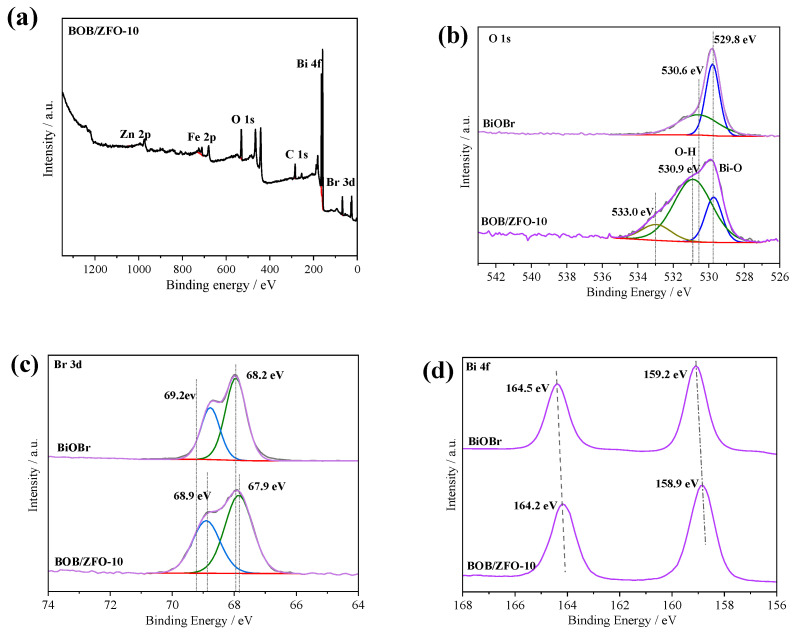
(**a**) XPS full spectrum of BOB/ZFO, BiOBr with (**b**) O1s, (**c**) Br3d, (**d**) Bi4f XPS spectrum of BOB/ZFO–10.

**Figure 6 nanomaterials-12-01508-f006:**
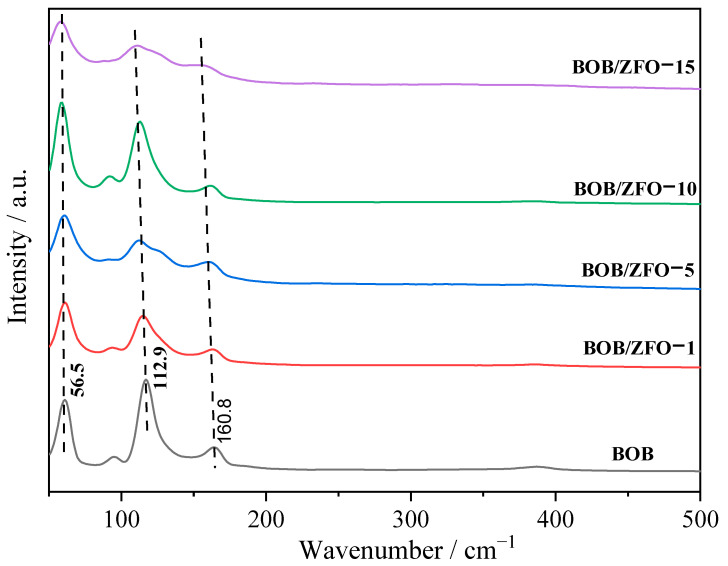
Raman spectra of the synthetic samples.

**Figure 7 nanomaterials-12-01508-f007:**
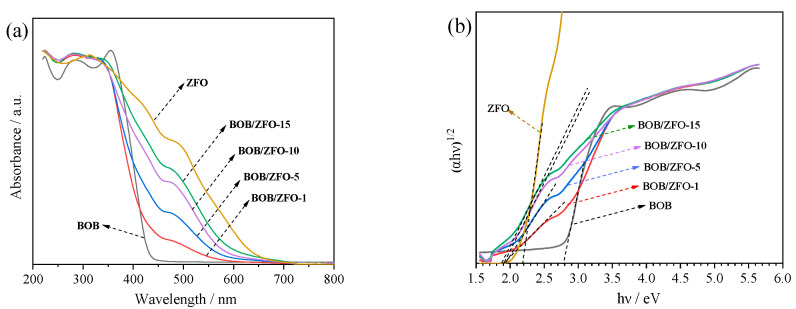
(**a**) UV-vis DRS spectra of the synthesized samples; (**b**) plots of (F(R∞)*hυ*)^1/2^ vs. *hυ* for the bandgap energies of the photocatalysts.

**Figure 8 nanomaterials-12-01508-f008:**
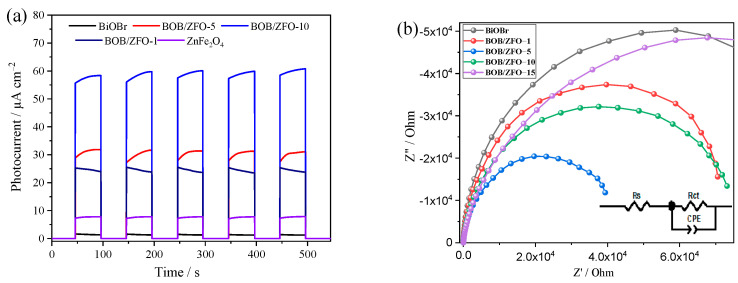
(**a**) The transient photocurrent response plots and (**b**) the electrochemical impedance spectra (EIS) for BiOBr, ZnFe_2_O_4_ and BOB/ZFO-x nanocomposites.

**Figure 9 nanomaterials-12-01508-f009:**
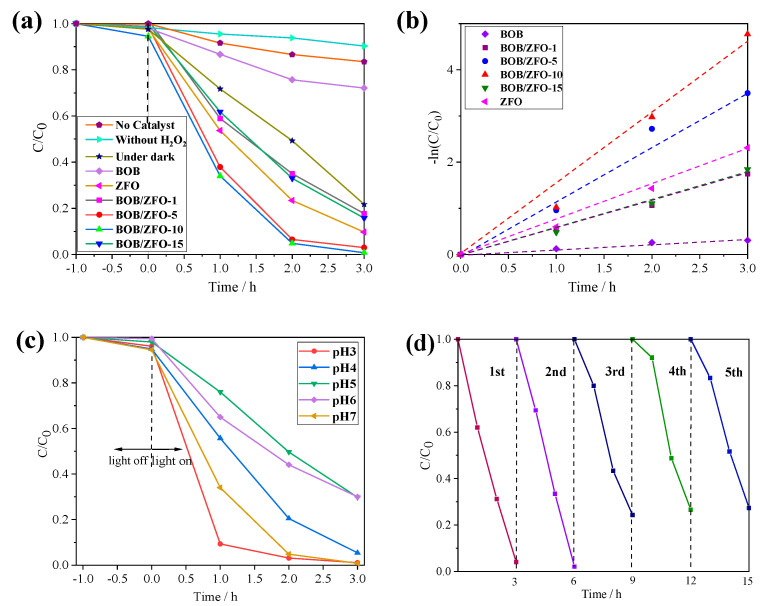
(**a**) TCH photodegradation activity over as-fabricated samples under visible light (λ > 420 nm); (**b**) the pseudo-first-order kinetic rate constants over all materials; (**c**) TCH photodegradation activity under various pH conditions (BOB/ZFO-10); (**d**) cycling degradation curves of TCH for BOB/ZFO-10 sample.

**Figure 10 nanomaterials-12-01508-f010:**
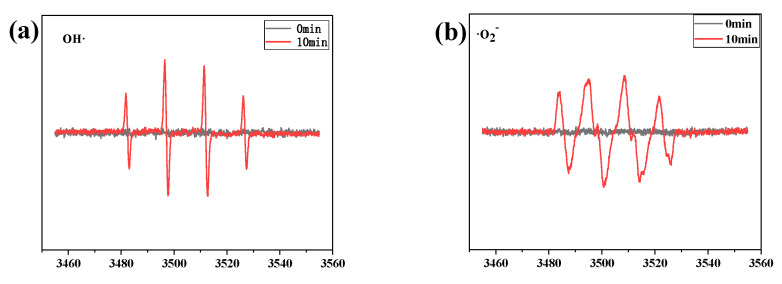
ESR signals of ˙OH (a) and ˙O_2_^−^ (b) measured using BOB/ZFO–10 as the sample.

**Figure 11 nanomaterials-12-01508-f011:**
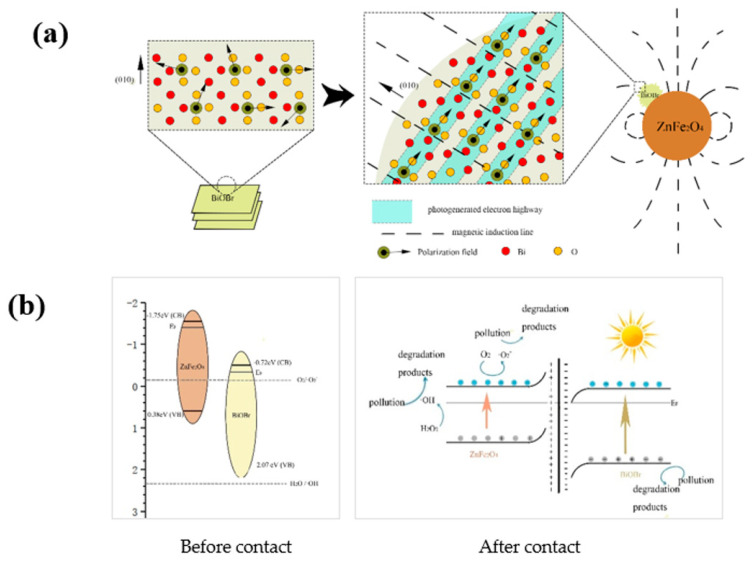
(**a**) The proposed different transportation route of photogenerated charges over pure BiOBr and BOB/ZFO composite (in which the BiOBr is epitaxial grown); (**b**) The band structure variation before and after the heterojunction formation and the enhanced mechanism of built-in electric field.

**Table 1 nanomaterials-12-01508-t001:** Specific surface area, pore volume and average pore size of BOB, ZFO and BOB/ZFO-x nanocomposites.

Material	Specific Surface Area (m^2^/g)	Pore Volume (mL/g)	Average Pore Size (nm)
BOB	6.98	0.037	9.9
BOB/ZFO-1	21.9	0.11	13.2
BOB/ZFO-5	22.4	0.11	11.3
BOB/ZFO-10	31.2	0.11	11.4
BOB/ZFO-15	24.2	0.10	12.9
ZFO	117.8	0.35	8.3
mechanically mixture (BOB/ZFO = 100:1)	8.09	-	-

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
