# Peer review of "Preparation, Characterization and Application of Epitaxial Grown BiOBr (110) Film on ZnFe2O4 Surface with Enhanced Photocatalytic Fenton Oxidation Properties"

_nanomaterials, 2022, doi:10.3390/nano12091508_

Round 1

Reviewer 1 Report

The paper needs some rectifications, clarifications and additions before one can take a final decision:

  1. The paper contains some grammatical errors and typo-mistakes that should be corrected. The English language should be improved.
  2. The Abstract part should be improved. Shortened it to be more clear to the reader. Abstract should clearly inform the important findings in the present study.
  3. The introduction part can be further improved. The authors are suggested to include recent references in the introduction part.
  4. The choice of the research topic is attractive. These are composites based on Bismuth oxybromide and spinel ferrite ZnFe2O4. These materials are very promising materials for several practical applications which can be highlighted in the Introduction part. However, in the introduction part, the authors did not sufficiently treat the importance/applications of the spinel ferrites. The introduction section needs to be improved by adding the current advancement in the magnetic nanomaterials for various applications.
  5. The authors should clarify the choice of Also, the motivation behind the present work with novelty statement should be added into the manuscript.
  6. How can the authors confirm the successful preparation of ZnFe2O4? The corresponding XRD pattern do not show anything.
  7. The structural properties could be further improved. What are the reasons for the shift in XRD peaks?
  8. The diverse structural parameters were NOT determined and discussed.
  9. The EDX spectra and chemical composition (atomic %) should be provided.
  10. The energy gap Eg is decreasing. What are the plausible reasons for such tendency? This should be clarified also in the text.
  11. The authors mentioned in the whole parts of the manuscript about “Ferroelectric ZnFe2O4”, but this was NOT confirmed in the paper as well as the ferroelectricity was NOT discussed. Accordingly, polarization measurements should be provided and investigated for the different systems.
  12. Photocatalytic results should be compared with those provided in the literature.

Author Response

Dear Respected Editors and Reviewers:

We are extremely appreciated with your email and comments concerning our manuscript entitled“Epitaxy Grown BiOBr (110) by Ferroelectric Effect of ZnFe2O4 with Enhanced Photocatalytic Fenton Oxidation Properties”. These comments are all valuable and very helpful for revising and improving the quality of our paper.

We have studied the comments carefully and have made corresponding correction which we hope would meet with approval. The corrections in paper and the responds to the reviewer’s comments are listed one by one as the following responses to the reviewer’s comments.

Best regards,

Yours sincerely

Prof. Yan Zhang

(On behalf of co-authors)

Reviewer 2 Report

  1. The title should be corrected as follows:

from

“Epitaxy Grown BiOBr (110) by Ferroelectric Effect of ZnFe2O4 2 with Enhanced Photocatalytic Fenton Oxidation Property”

to

“Epitaxy Grown BiOBr (110) by Ferroelectric Effect of ZnFe2O4 2 with Enhanced Photocatalytic Fenton Oxidation Properties”

  1. The authors should improve the English text and rewrite Abstract, Introduction, Results and Conclusions.
  2. BET surface area and pore size distribution results should be added if possible.
  3. The authors should correct the references according to journal requirements.

Author Response

(The authors gave the same response as above.)

Reviewer 3 Report

In this paper the authors describe the fabrication of BiOBr nanopalates on ZnFe2O4 and the utilization of these materials in the photocatalytic degradation of tetracycline antibiotics. The preparation was performed by solvothermal technique in which precursors for generating the BiOBr were added to ethylene glycol containing ZnFe2O4. The resulting materials are well characterized by different methods and structure elucidation of such materials was performed elegantly. The authors showed that the highest photocatalytic activity was obtained at pH 3 or 7, while at other acidic pH values the activity is decreased. There is no explanation for this catalytic behavior and the authors should add this to the text. Also, what is the stability of the materials at acidic pH? This should be analyzed and the materials after photocatalytic reactions should be analyzed to determine there are no changes in their structure.

Author Response

(The authors gave the same response as above.)

Round 2

Reviewer 1 Report

The paper has been improved. The authors tried to perform the required corrections. I think that it can be now accepted for publication in Nanomaterials-MDPI.

Author Response

Thank you for accepting the manuscript.

Reviewer 2 Report

Review of the paper MDPI Nanomaterials-1666382 R1

I recommend acceptance of the paper after minor revision.

Some of my recommendations 7 days ago were not fulfilled in the R1 manuscript. Please correct this.

Meanwhile I have some new questions notes and recommendations.

All new and unfulfilled old recommendations are written below.

  1. The authors should correct English text and re-write the Title, Abstract, Introduction, Results and Conclusions.

Title

“Epitaxy Grown BiOBr (110) by Ferroelectric Effect of ZnFe2O4 with Enhanced Photocatalytic Fenton Oxidation Properties”

Should be corrected as following:

“Preparation, ,Characterization and Application of Epitaxial Grown BiOBr (110) Film on ZnFe2O4 Surface with Enhanced Photocatalytic Fenton Oxidation Properties”

  1. Abstract

“A novel BiOBr photocatalyst was epitaxy grown in situ onto the surface of ZnFe2O4, ferroelectric material with strong polarization effect”

Should be replaced with:

“A novel BiOBr photocatalyst was epitaxially grown in situ onto the surface of ZnFe2O4, a ferroelectric material with strong polarization effect”

  1. Introduction

Page 1, Row 39-40

“At present, the common treatment processes of wastewater include physical method, 39 chemical method, physicochemical method and biological method”

Should be substituted by:

“In our days the main wastewater treatment processes are based on physical, chemical, physicochemical and biological methods”

4.

“2.2. Characterizations”

Should be replaced with:

“2.2. Methods of Characterization:

5.

“3. Results and Discussions”

Should be substituted by:

 “3. Results and Discussion”

6.

“Figure 2. SEM photograph of (a) ZnFe2O4, (b) BiOBr, (c) BOB/ZFO-5, (d) BOB/ZFO-15”

Should be replaced with:

“Figure 2. SEM photographs of (a) ZnFe2O4, (b) BiOBr, (c) BOB/ZFO-5, (d) BOB/ZFO-15”

7.

“Figure 7. (a) UV-vis DRS spectra of the synthesized sample”

Should be substituted by:

“Figure 7. (a) UV-vis DRS spectra of the synthesized samples”

8.

“3.6 The photoelectrochemical property of BOB/ZFO nanocomposite”

Should be replaced with:

“3.6 The photoelectrochemical properties of BOB/ZFO nanocomposites”

  1. Authors should correct the reference list according to journal requirements (Ref. 35, etc.).

10.

“Conclusion”

Should be substituted by:

“Conclusions”

  1. Conclusions

Page 15, rows 560-564

“In summary, the BiOBr/ZnFe2O4 nanocomposite photocatalysts with an epitaxy grown BiOBr(110) crystalline facet have been synthesized by the ferroelectric polarization effect ZnFe2O4. The BOB/ZFO-10 composites exhibited the highest photocatalytic degradation performance for tetracycline hydrochloride under visible-light irradiation for 3 h, 563 achieved 99.2% degradation, which is 3.56 times efficient than that over pure BiOBr.”

Should be replaced with:

“In summary, the BiOBr/ZnFe2O4 nanocomposite photocatalysts have been synthesized by epitaxial growth of BiOBr(110) crystalline facet on ferroelectric ZnFe2O4 surface. The BOB/ZFO-10 composites exhibit the most efficient photocatalytic degradation performance for tetracycline hydrochloride under visible-light irradiation for 3 h - 99.2% degradation degree, which is 3.56 times higher than that over pure BiOBr.”

Author Response

Thank you very much for your careful check and good suggestions. We are very sorry for the grammatical errors and typo-mistake that we have made. Now we have revised all of the comments above according to your suggestions. The corresponding changes have also been made in the manuscript.
